# The Immunometabolic Atlas: A tool for design and interpretation of metabolomics studies in immunology

**Pascal Maas**[ID][☯], **Ilona den Hartog**[ID][☯], **Alida Kindt, Sonja Boman**[ID]**, Thomas Hankemeier, Coen van Hasselt** *

Division of Systems Biomedicine & Pharmacology, Leiden Academic Centre for Drug Research, Leiden University, Leiden, The Netherlands

☯ These authors contributed equally to this work.
* coen.vanhasselt@lacdr.leidenuniv.nl

**Data Availability Statement:** All relevant data are within the manuscript and its Supporting Information files.

## Abstract

Immunometabolism, which concerns the interplay between metabolism and the immune system, is increasingly recognized as a potential source of novel drug targets and biomarkers. In this context, the use of metabolomics to identify metabolic characteristics associated with specific functional immune response processes is of value. Currently, there is a lack of tools to determine known associations between metabolites and immune processes. Consequently, interpretation of metabolites in metabolomics studies in terms of their role in the immune system, or selection of the most relevant metabolite classes to include in metabolomics studies, is challenging. Here, we describe the Immunometabolic Atlas (IMA), a public web application and library of R functions to infer immune processes associated with specific metabolites and vice versa. The IMA derives metabolite-immune process associations utilizing a protein-metabolite network analysis algorithm that associates immune system-associated annotated proteins in Gene Ontology to metabolites. We evaluated IMA inferred metabolite-immune system associations using a text mining strategy, identifying substantial overlap, but also demonstrating a significant chemical space of immune system-associated metabolites that should be confirmed experimentally. Overall, the IMA facilitates the interpretation and design of immunometabolomics studies by the association of metabolites to specific immune processes.

## Introduction

Immunometabolism, or the interplay of immunology and metabolism, has received increasing interest because of its role in the function and regulation of immune system processes in health and disease. Metabolites with *e.g.*, pro- or anti-inflammatory functions may be of interest as biomarkers or drug targets for inflammation and immune system-associated pathologies such as infection, cancer, and various auto-immune diseases [1–3]. Significant knowledge gaps related to the relationship between metabolism and immune function remain to be elucidated. To this end, metabolomics technologies can facilitate the identification and quantification of metabolites in relation to the immune system in experimental models and clinical studies.

**Funding:** This work is part of the research program 'Metabolomic fingerprint biomarkers to guide antibiotic therapy and reduce resistance' with project number 541001007, which is financed by ZonMW, the Netherlands Organization for Health Research and Development associated with the Dutch Research Council (NWO). The funders had no role in study design, data collection and analysis, decision to publish, or preparation of the manuscript.

**Competing interests:** The authors have declared that no competing interests exist.

For biochemical and functional interpretation of metabolomics study results, different computational tools can be used: biochemical pathways analysis can be executed using tools such as MetaboAnalyst or KEGG, and for functional analysis, STITCH can be employed [4–6]. However, inferring the relationship of metabolites with immune system processes remains challenging. In contrast, for the analysis of genes, gene expression, and proteins, such biological interpretation is straightforward through the use of high-quality annotated ontologies such as Gene Ontology [7, 8].

For hypothesis-driven metabolomics studies that require absolute quantification of measured metabolites, targeted metabolomics methods are preferred over untargeted metabolomics methods. However, targeted mass spectrometry-based metabolomics studies measure by design only a subset of metabolites and metabolite classes at once. Guidance in the selection of the most relevant subset of metabolites for the immune process of interest is therefore of relevance. However, tools to facilitate the design of targeted metabolomics studies by pre-selection of metabolites of interest are lacking.

To address the current hurdles of hypothesis generation and biological interpretation of metabolomics studies, we developed the Immunometabolic Atlas (IMA). The IMA enables inference of immune system associated functions, and vice versa, to determine relevant metabolites with specific immune system processes. We infer metabolite-immune process associations utilizing a protein-metabolite network analysis algorithm that associates immune system-associated annotated proteins (Fig 1), leveraging protein-metabolite interaction databases [9, 10] and protein annotations of immune system processes in Gene Ontology (GO). We then characterize the global metabolite-immune process coverage and perform validation through text mining-derived immune system associations. The application of the IMA is demonstrated in a case study and is made available as an R package and public web application.

## Methods

### Assembly of immune process-metabolite interaction network

We constructed a database that contains associations between specific immune process terms, proteins, and metabolites through the integration of publicly available databases (Fig 2A).

## The Immunometabolic Atlas

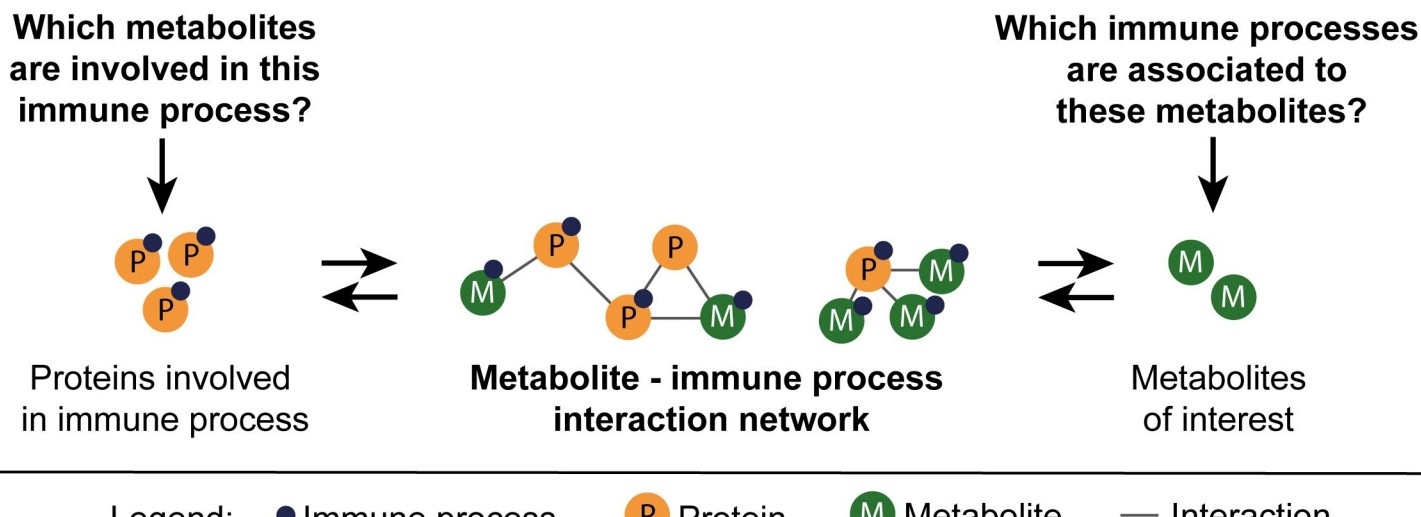

**Fig 1. A conceptual overview of the Immunometabolic Atlas (IMA).** The IMA provides associations between metabolites and immune processes of interest through the generation and evaluation of a protein-metabolite interaction network.

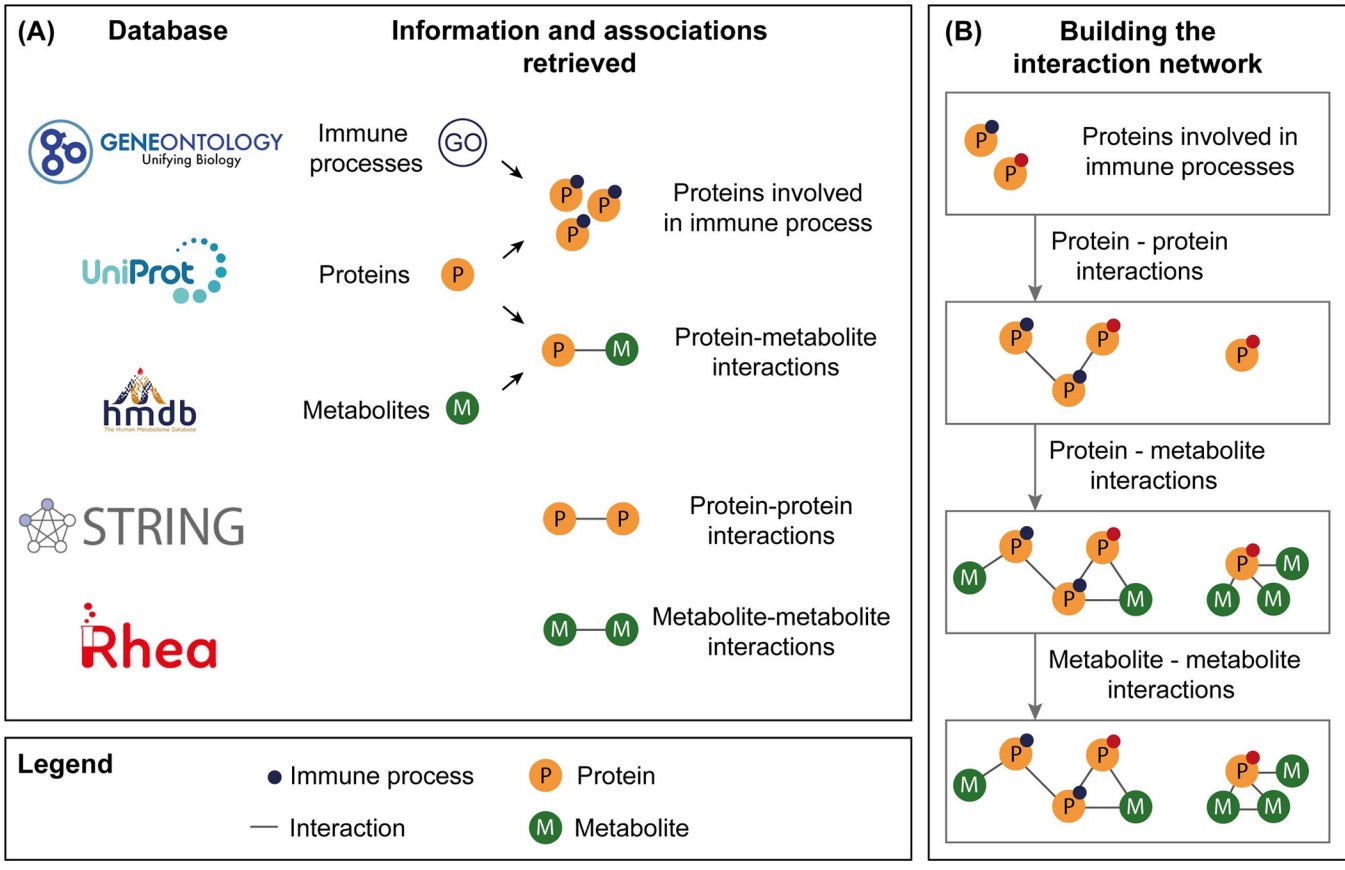

**Fig 2.** Overview of (A) information and associations retrieved from available databases, and (B) the study flow to build protein-metabolite interaction networks to associate metabolites and immune processes of interest. First, the proteins that are associated with the immune processes of interest are added to the network. Then, protein-protein, protein-metabolite, and metabolite-metabolite interactions are added to the network to generate the final interaction network for the immune processes of interest.

Through the integration of these resources, we constructed an interaction network to associate metabolites with immune processes. In the following paragraphs, the development of the immune process-metabolite interaction network is described.

**Immune processes.** Immune processes were retrieved as GO terms from Gene Ontology. The associated gene names that were descendants of "Immune System Process" (GO:0002376) were acquired using the EBI QuickGo application programming interface (API, version 2021-05-24) [7, 8, 11].

**Proteins and protein-immune process associations.** Human proteins (Swiss-Prot) were retrieved from the UniProt database [12]. The requested UniProt data included: entry (UniProt identifiers), protein name, cofactors, EC number, transporter protein (TCDB), Ensembl transcript, and GO immune processes. The reference to GO immune processes in the UniProt data was used to identify immune system-related proteins.

**Metabolites.** Metabolite names and associated metadata were obtained from the Human Metabolome Database (HMDB, version 4.0). We only included metabolites that were known to either have a biological role and/or were part of a naturally occurring process to exclude any synthetic drugs. We also excluded any inorganic compounds. The retrieved HMDB data included: name, class, superclass, accession (HMDB identifiers), ChEBI ID, UniProt ID, biospecimen, cellular locations, and metabolic pathways.

**Protein-protein interactions.**   Protein-protein interactions were obtained from STRING's functional protein association networks version 11.0 [9]. Ensembl transcripts from the Uni-Prot data were converted to Ensembl Protein IDs using the Ensembl API [13]. Subsequently, STRING was parsed using these IDs to extract protein-protein interactions.

**Metabolite-protein interactions.**   The UniProt identifiers in the HMDB data were used to connect the metabolites to the proteins in the UniProt data, obtaining metabolite-protein interactions. Proteins without immune system-related GO terms were excluded from further analysis.

**Metabolite-metabolite interactions.**   Metabolite-metabolite interactions for the obtained metabolites from HMDB were retrieved using the Rhea-Annotated reactions database (RheaDB, release 118) [10]. We cross-referenced HMDB with ChEBI to extract interactions stored in Rhea. We applied an all-versus-all method, where each reactant-product combination results in an individual interaction.

**Building the interaction network.**   To construct the interaction network for each immune process extracted from GO, first, proteins involved in the immune processes were identified. Then, protein-metabolite, protein-protein, and metabolite-metabolite interactions were added to the network. To build an interaction network for metabolites of interest, proteins associated with the metabolites of interest were identified. Related protein-metabolite, protein-protein, and metabolite-metabolite interactions were then added to the network (Fig 2B). For metabolites with only metabolite-metabolite interactions, no interaction network can be constructed, because at least one protein-metabolite interaction is necessary to inherit immune processes.

**Inheritance of immune processes by metabolites.**   To associate metabolites to immune processes, an inheritance methodology was applied (Fig 3B). In the default, first-order inheritance method, metabolites inherit the immune processes of the directly neighboring proteins only. For second-order and third-order inheritance, metabolites inherit both the immune processes of their direct neighboring proteins and the first neighbors of that protein, two or three interaction steps away, respectively. The preferred inheritance order can be defined by the user.

## Evaluation of network-inferred metabolite and immune processes

**Overrepresentation analysis.**   We test for the overrepresentation of metabolites and immune processes in the interaction network using Fisher's exact test with multiple testing correction, using the IMA metabolites and immune processes as background. Based on this, we rank by p-value to identify the most significant metabolites or immune processes associated with either an immune process or metabolite set. The p-value for Fisher's exact test was computed as follows (Eq 1):

$$p - value\ Fisher's\ exact\ test = \frac{(a+b)!(c+d)!(a+c)!(b+d)!}{a!b!c!d!(a+b+c+d)!} \tag{1}$$

Here, for metabolite-based overrepresentation analysis, $a$ is the number of associations of a specific metabolite to a specific immune process in the interaction network (via multiple proteins), $b$ is the number of associations of other immune processes to the specific metabolite in the network, $c$ is the total number of associations of the specific metabolite to the specific immune process in the database minus the number of associations of the specific metabolite to the specific immune process in the network, and $d$ is the total number of immune process associations to the specific metabolite in the database minus the number of associations of other immune processes to the specific metabolite in the network. For immune process-based overrepresentation analysis, $a$ is the number of appearances of a specific immune process in

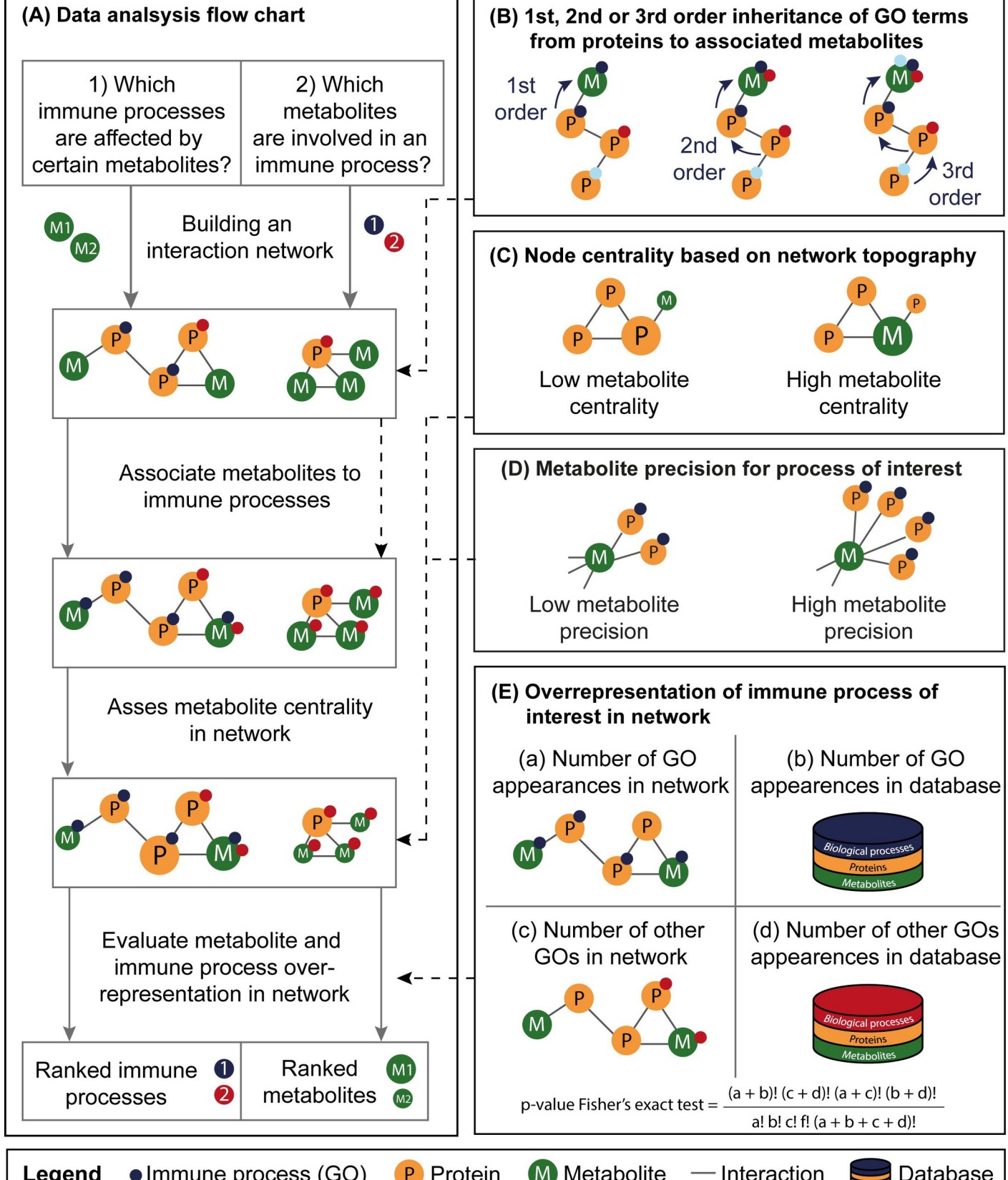

**Fig 3. Overview of the IMA network-based interaction analysis.** (A) Flowchart of the data analysis of an interaction network using a first-order inheritance strategy. (B) The metabolite of interest can inherit immune processes directly (1st order, default) or indirectly (2nd or 3rd order) from neighboring proteins based on the order of inheritance chosen by the user. (C) The centrality of a protein or metabolite in a graphical network is determined using the harmonic closeness score. This topology-based score is the highest for metabolites with multiple connections in a central point in the network. (D) The precision of a metabolite for a process of interest is determined by the ratio of its interactions within the process compared to all its interactions and represents the commitment of a metabolite to the process of interest. (E) The overrepresentation of an immune process (GO term) in an interaction network is evaluated using a Fisher's Exact Test with FDR multiple testing correction. The resulting significance levels can be used to rank immune processes in a network.

the interaction network, $b$ is the number of the immune process appearances in the IMA database, $c$ is the number of other immune processes in the network, and $d$ is the number of other immune processes appearances in the IMA database.

**Metabolite centrality.** We calculated metabolite centrality to determine the position of a metabolite in the interaction network. A metabolite could be on the edge of a network with minimal interactions, in the center of a network with a lot of interactions, or somewhere in between. The centrality was calculated as harmonic closeness, which is a distance-based centrality metric that is suitable for disconnected graphs, in contrast to classical closeness. A high harmonic closeness value indicates a central position of the metabolite in the network. For node $i$, the harmonic closeness is calculated by taking the sum of all reciprocals of distance $d$ to other node $j$ (Eq 2). The centrality was determined for each metabolite in the network separately.

$$Harmonic\ closeness(i)\ \sum_{j \neq i} \frac{1}{d_{i,j}} \tag{2}$$

**Metabolite precision.** Metabolite precision was computed to quantify how specific a metabolite is for a certain immune process. The precision score was computed for each metabolite-immune process association to allow discrimination between metabolites that are contributing either to a single process or to multiple processes and between common and rare metabolites that have comparable centrality scores. The precision of a metabolite for an immune process of interest is determined by the ratio of the metabolite associations with the process of interest, compared to all its associations remaining in the IMA database (Eq 3). A high metabolite precision value indicates that most of the immune process associations the metabolite could have, according to the IMA database, are present in the immune process network.

$$Precision\ score(i,j) = \frac{N_{i,j}}{N_i \cdot V_j} \tag{3}$$

Here, for metabolite $i$ in process $j$, with $N$ being the number of interacting nodes of metabolite $i$ and $V$ the number of nodes in process $j$. The precision score is corrected by the number of nodes in the process.

## Evaluation of IMA metabolite-immune process association performance through text mining

To evaluate the evidence available for metabolite-immune process associations identified by the IMA, an external validation dataset was created using text mining. We selected papers including one or more metabolites and immune processes that were also present in the IMA database using the EuropePMC API on 6 March 2021 [14]. We included EuropePMC-listed journal articles in which a metabolite and immune process term from the IMA database was

detected in the abstract, methods, results, supplement, figures, and/or tables. Introduction and discussion sections were excluded since comparisons to results of other studies are often made in these sections, possibly leading to biased text mining results. Also, papers were only included if they were related to humans.

The text mining resulted in a list of PubMed identifiers (PMIDs) which were used to find associations between metabolites and immune processes. These associations were included in the quantitative text mining validation dataset. Metabolite-immune process associations with only one occurrence in the text mining dataset were removed to limit false positives. We excluded the superclass lipids and lipid-like molecules as defined within HMDB from the validation because the complex nomenclature of these metabolites made text mining unfeasible.

We characterized the IMA database by cross-referencing metabolites and processes with the text mining database. Metabolites were grouped according to their presence or absence in the IMA database. Furthermore, we evaluated the quality of the IMA database by calculating the specificity, sensitivity, precision, accuracy, and $F_1$-score (Eqs 4–8). The $F_1$-score focuses on the positive predictions and leaves out the True Negatives. The $F_1$-score represents the performance of the IMA better than other evaluation measures because it evaluates how well associations are made instead of how well associations are excluded.

$$Specificity = \frac{True\ Negatives}{True\ Negatives + False\ Positives} \tag{4}$$

$$Sensitivity = \frac{True\ Positives}{True\ Positives + False\ Negatives} \tag{5}$$

$$Precision = \frac{True\ Positives}{True\ Positives + False\ Positives} \tag{6}$$

$$Accuracy = \frac{True\ Negatives + True\ Positives}{True\ Negatives + True\ Positives + False\ Negatives + False\ Positives} \tag{7}$$

$$F_1 - score = \frac{True\ Positives}{True\ Positives + \frac{1}{2}(False\ Positives + False\ Negatives)} \tag{8}$$

## R package and Shiny application

We implemented the IMA in the R package IMatlas, which facilitates users to create various graph-based analyses. The package includes an interactive R shiny application that allows for a user-friendly interpretation of our interaction database. The app adds extensions that are useful for additional analyses, including metadata from HMDB and UniProt, and allows networks to be built using either one or multiple immune processes, or by one or multiple metabolites. If one or multiple immune processes are used as input, all connected metabolites that are in the Immunometabolic Atlas database will be included in the graphical network. The app also features two additional versions of interaction datasets, which allows users to determine the strictness of the app. These datasets include proteins that are unrelated to the immune system but do have an interaction with an immune system-related protein. The first dataset includes neighbors of immune system-related proteins, whereas the second dataset includes the second neighbors of an immune system-related protein. The package and all other scripts used for analysis are available in our Github repository https://github.com/vanhasseltlab/IMatlas.

# Results

## Development of the IMA database and metabolite-immune process algorithm

The IMA database includes all child processes of the immune system process (GO:0002376) and contains 97 525 metabolites, 3 101 proteins, 1 712 immune processes, 664 metabolite-metabolite interactions, 172 291 protein-metabolite interactions, and 411 286 protein-protein interactions (S1 Table).

We associated immune processes and metabolites in a stepwise process (Fig 3). Immune processes were assigned to metabolites using a first, second, or third-order inheritance strategy (Fig 3B). By default, first-order inheritance of immune processes is used, in which metabolites only inherit immune processes from their directly interacting protein. To determine if a metabolite of interest plays a central role in the metabolite-immune process interaction network, a centrality score was calculated (Fig 3C). Metabolites with a high centrality score are typically located in a central point in the network and have multiple interactions with surrounding metabolites and proteins, while metabolites with a low centrality score are less closely connected to other metabolites or proteins in the network and are typically located towards the edges of a network. To indicate how specific a metabolite is for a certain immune process, the precision score was calculated (Fig 3D). The precision of a metabolite for an immune process of interest is determined by the ratio of the metabolite associations with the process of interest, compared with all its associations remaining in the database. Metabolites with a high precision score are typically committed to a smaller number of immune processes. To rank metabolites and immune processes in the interaction network, we calculated a p-value that signifies the overrepresentation of the metabolites and immune processes in the network in comparison to the ones in the database using Fisher's exact test (Fig 3E). This resulted in a performance table with significance values for every metabolite and immune process within the network. The significance value for overrepresentation of the immune process for a specific metabolite is indicative of the strength of metabolite-immune process association. The network-based significance value that indicates the overrepresentation of an immune process within the entire network indicates the importance of the collection of metabolites for the immune process. In summary, the metabolites and immune processes in the network are ranked based on their metabolite centrality, precision and p-value, and the immune process p-value (Fig 3A).

## Overview of metabolism-immune response associations

To provide an overview of IMA-inferred metabolite-immune response process associations, we categorized GO terms according to main high-level immune response processes as defined in the standard textbook Janeway's Immunobiology [15]. We determined for each of these immune response processes the biochemical metabolite superclasses of identified metabolites (Fig 4). We found significant differences in metabolite classes associated with unique immune response processes (Fig 4A). The average number of immune processes per protein is in the same order of magnitude for all superclasses except for benzenoids (S2 Table). Metabolites of the superclass of lipids and lipid-like molecules, here referred to as 'lipids', were abundantly present with 90 280 occurrences (92.6%) but interacted with a relatively small portion of proteins (5.1%). Excluding lipids from the analysis shows a lower average number of metabolites that were associated with a specific immune process. There were 18,995 unique metabolites associated with the main immune processes when lipids were included and 342 when they were excluded. A large variation of the number of metabolites associated with immune processes was present (Fig 4B). Excluding lipids results in a shift from many metabolites to smaller

**(A)**

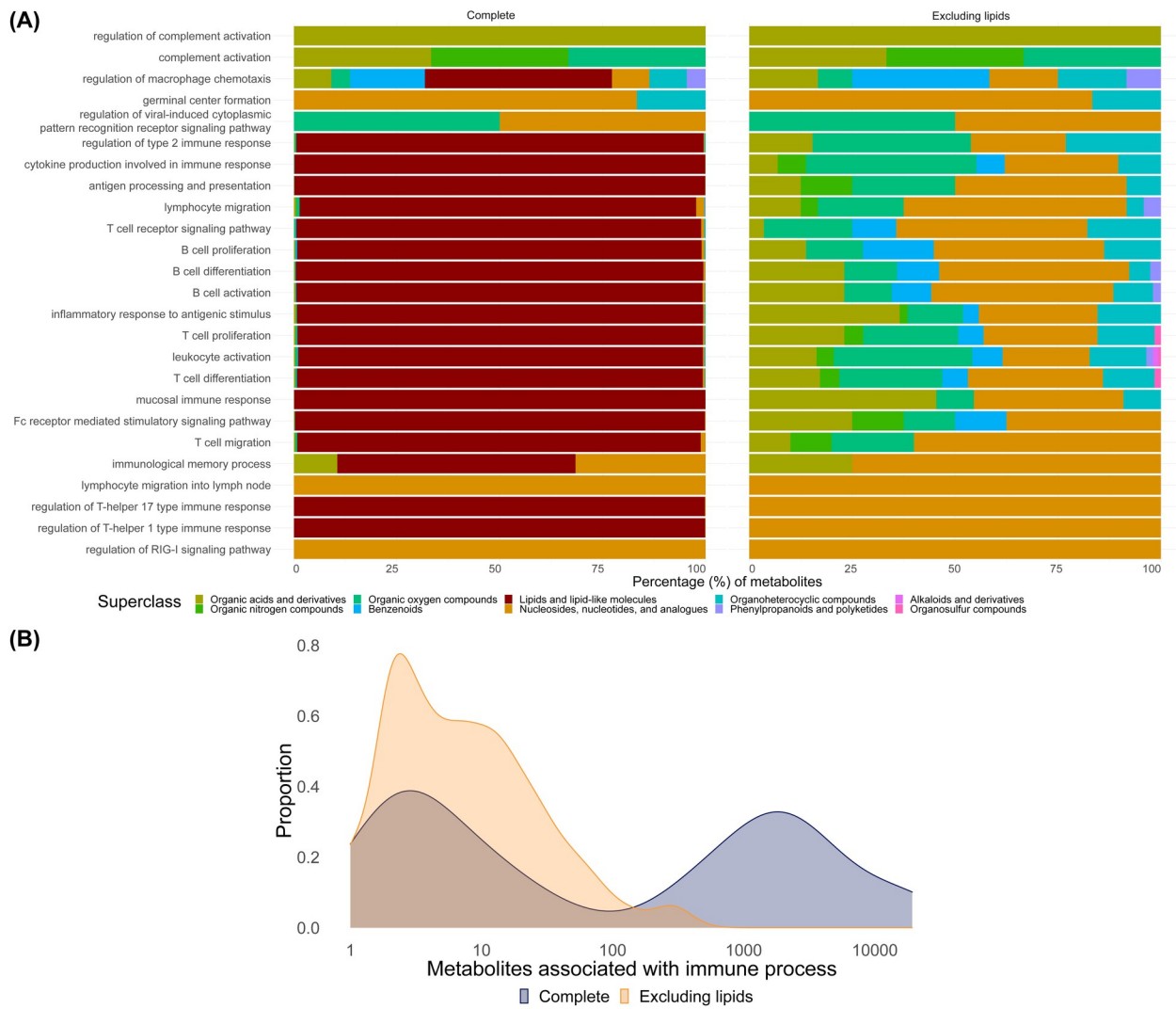

**Fig 4. Overview of immune process and metabolite associations.** (A) Distribution of the biochemical metabolite classes identified for common immune processes in the IMA classified according to the standard textbook Janeway's Immunobiology, either including (left) or excluding (right) lipids. (B) Distribution of the number of metabolites associated with specific immune processes inferred from the IMA, using first-order inheritance, either for excluding lipids (orange) or including all metabolites including lipids (blue).

numbers of metabolites that are associated with an immune process. The exclusion of the superclass of lipids and lipid-like molecules from these results excludes several metabolite classes including fatty acyls, glycero(phospho)lipids, and prenol lipids.

## Validation of the metabolite-immune process associations

The methodology was validated by comparing the results from an interaction analysis of all metabolites and immune processes in the IMA database to metabolite-immune process associations found in literature for the same metabolites and immune processes. The immune system process interaction network was built using 1st order immune process inheritance and resulted in 432 metabolites associated with 767 immune processes.

Associations of metabolites and immune processes related to the immune system process in literature were collected using a text mining approach. We identified 1 046 metabolites that were

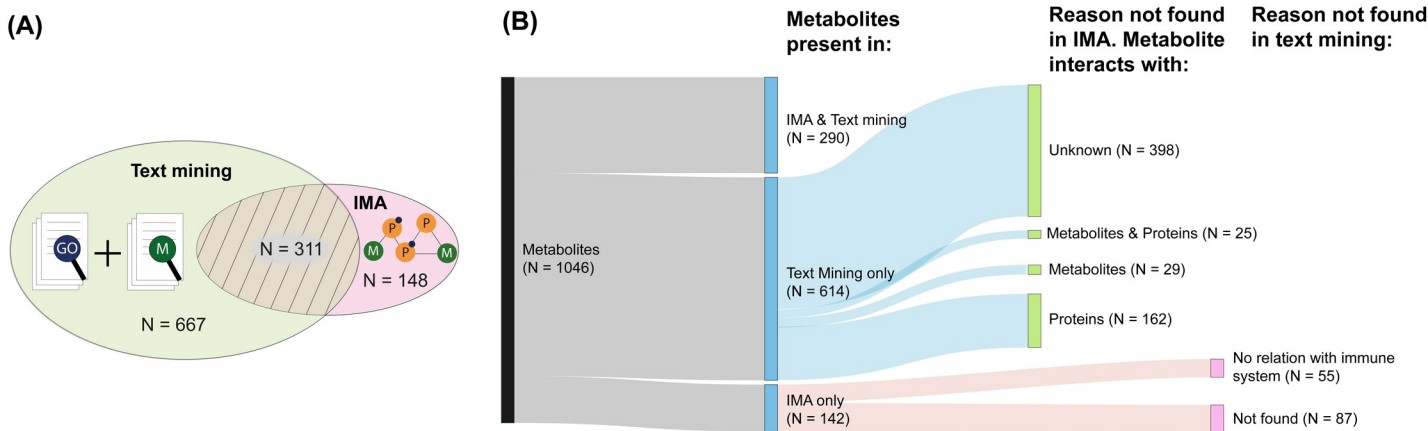

**Fig 5. Comparison of metabolites in IMA database and text mining dataset for validation.** (A) Text mining was used to identify co-occurrences of GO-terms and metabolites present in the IMA database. Metabolites obtained from co-occurrences were compared with associations found using IMA. (B) Sankey diagram of metabolites found in both IMA and text mining. 290 of 1 046 metabolites found are present in both IMA and text mining datasets. The portion of non-overlapping metabolites can be explained based on exclusion criteria for the atlas. 614 metabolites were only found in literature, of which 398 have no known interaction to any proteins or metabolites in the IMA. There were interactions found of the metabolites to other metabolites (n = 81), metabolites and proteins (42) or proteins (160), but these did not have an association to an immune process in the IMA database. Of the 142 metabolites that were only found in the atlas, 87 could not be found in literature and 55 were found in literature but missed any connection to an immune process.

associated with 565 immune system processes (S3 Table). The overlap between associations found by the IMA and found in literature was 31.5% (290 metabolites in 398 processes, Fig 5).

Of all associations found in text mining and by the IMA, 58.7% of the metabolites involved in metabolite-immune process associations were only found in literature and not by the IMA (n = 614 metabolites). Of these, 398 metabolites were lacking any interaction according to the IMA database and therefore remained undetected using the IMA methodology. The 216 remaining metabolites with known interactions could be classified as having either only metabolite-metabolite interactions and/or protein-protein interactions. Metabolites that were only interacting with other metabolites, and not with proteins, could not be detected because immune processes are only inherited through proteins in the current IMA methodology. Metabolites that were only interacting with proteins that were not in the immune system process (according to GO), were also not included in the IMA database. Finally, 13.6% of the metabolites that inherited an immune system process were only found using the IMA and not in literature (n = 142). Of these, 55 metabolites were identified in literature but were lacking a link to the immune system. The remaining 87 metabolites were not found in any immune-related studies through text mining.

The metabolite-immune process associations found in literature were considered as the gold standard for the evaluation of the performance of different orders of immune process inheritance. By default, the inheritance of processes was done through direct protein interactions (first-order), but inheritance through indirect protein interactions was also evaluated (second and third-order). Therefore, specificity, sensitivity, precision, $F_1$-score, and accuracy were calculated (Table 1). All methods of inheritance yielded high specificity and accuracy

**Table 1. Absolute performance measure results of first, second, and third-order inheritance.**

| Order of inheritance | Specificity | Sensitivity | Precision | Accuracy | $F_1$-score |
|---|---|---|---|---|---|
| First order | 0.99 | 0.11 | 0.29 | 0.97 | 0.16 |
| Second order | 0.85 | 0.43 | 0.07 | 0.84 | 0.12 |
| Third order | 0.73 | 0.49 | 0.04 | 0.73 | 0.08 |

values, indicating that the IMA is strict in linking processes to metabolites. Relatively low values for precision and sensitivity were reported, indicating discrepancies between the associations found in literature and made by the IMA. Comparing direct- and indirect inheritance showed a higher precision for direct inheritance, while indirect inheritance showed higher sensitivity. Since the IMA gives high numbers of True Negatives, also the $F_1$-score, which does not take the number of true negatives into account, was calculated to quantify the difference between the methods of inheritance. The $F_1$-score favoured direct inheritance.

## Identification of possible biomarkers using network-based interaction analysis

To identify which metabolites could be of interest for a specific immune process, we reported the position of a metabolite in the network (centrality) and the exclusivity of the metabolite for a certain immune process (precision, Fig 6B and 6C). Metabolites with high centrality and precision scores might be of interest as biomarkers for the associated immune process. Therefore, all metabolite-immune process associations made by the IMA were analyzed on centrality and precision. Only statistically significant metabolite-immune process associations after FDR

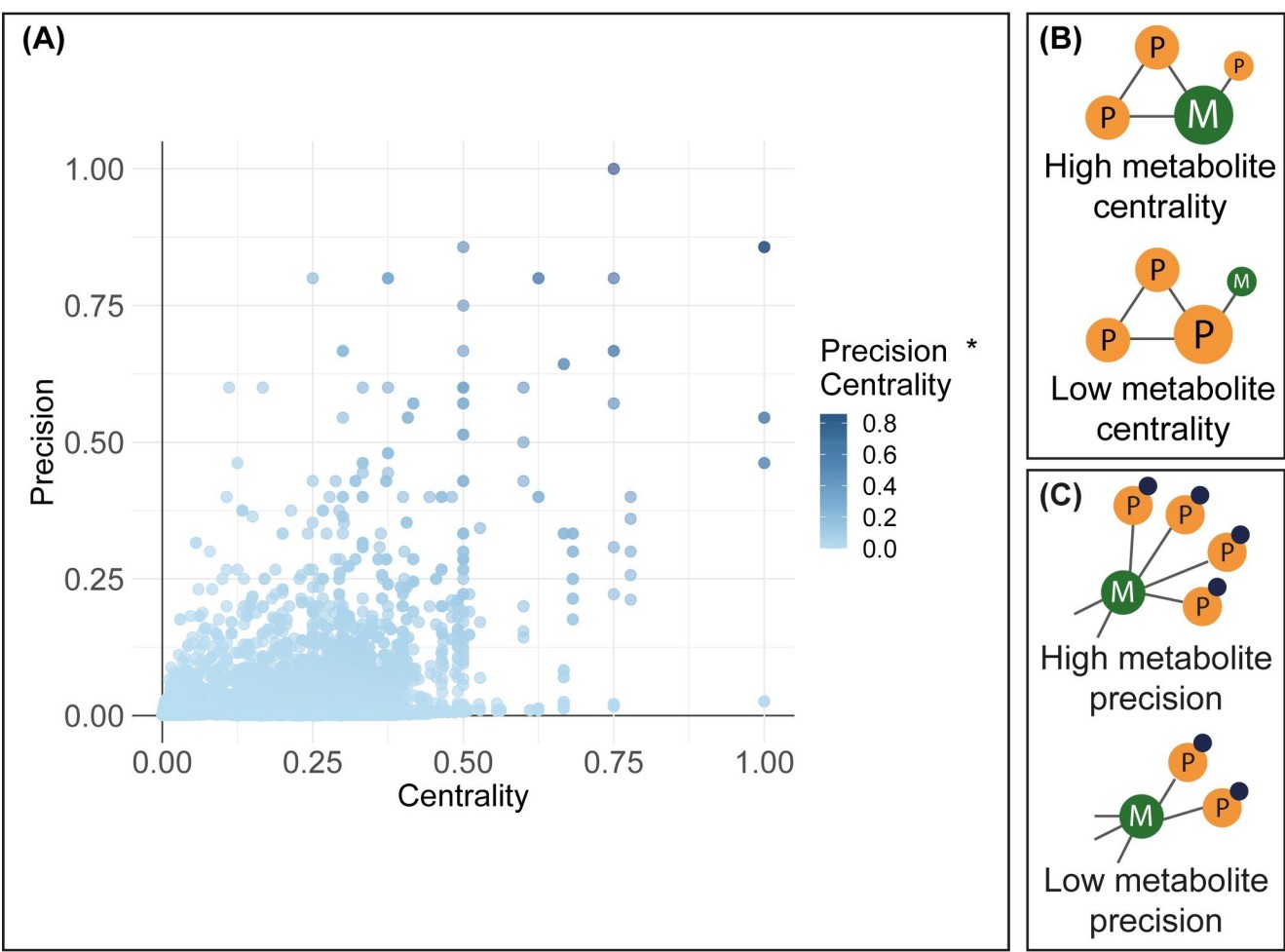

**Fig 6. Precision-centrality plot of the significant metabolite-immune process associations in the IMA excluding lipids.** (A) For each metabolite in each immune process, the centrality and precision were calculated and normalized to the network size. Metabolites with high centrality and precision scores might be of interest as biomarkers for the associated immune process (B) The centrality represents the position of a metabolite in the network. (C) The precision represents the exclusivity of the metabolite for a certain immune process.

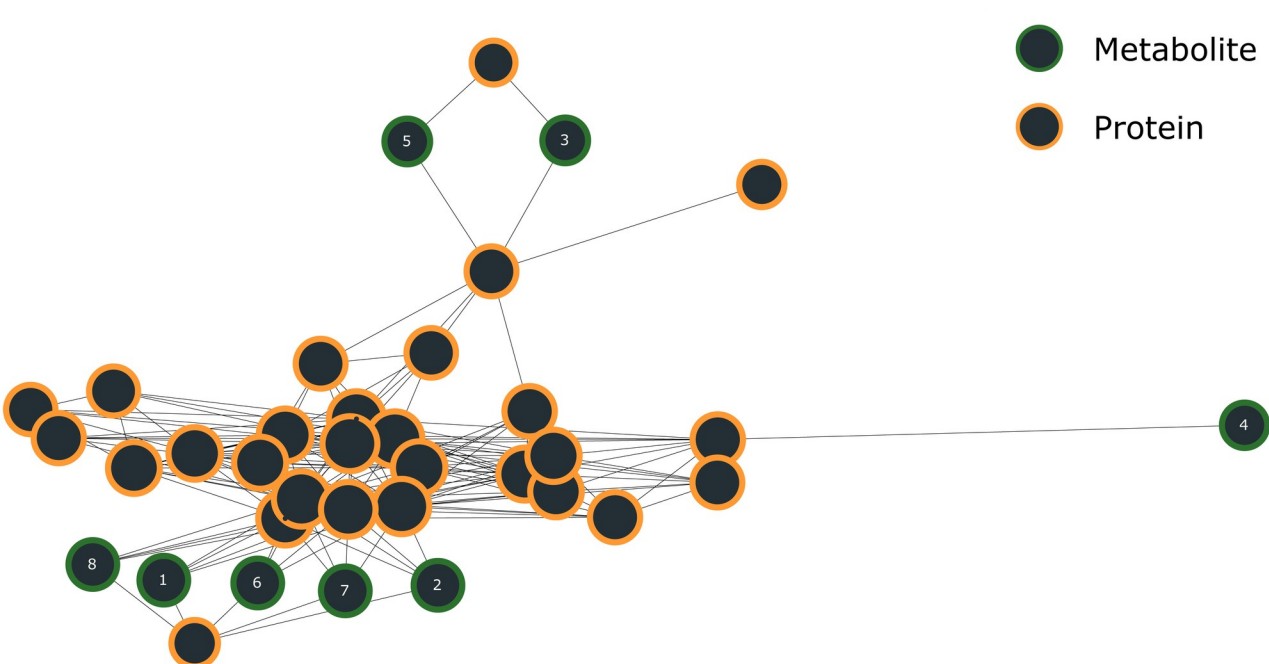

**Fig 7.** The interaction network of positive regulation of T cell-mediated immunity (GO:0002711) shows the interplay between metabolites (green) and proteins (orange).

multiple testing correction were included in the results (p < 0.05, Fig 6A, S4 Table). To identify specific metabolic biomarkers, we selected metabolite-immune process associations that were above the set threshold of the mean plus two standard deviations for both centrality and precision. After analysis of the associations in the current IMA association dataset, 48 metabolites were found to be of interest as a potential biomarker in 47 immune processes. Several metabolites were found to be involved in antigen processing via MHC class 1B, including sphingosine, sphinganine, and dihydroceramide. Furthermore, we identified strong relationships between several pyruvic acids and positive regulation of prostaglandin secretion.

### Positive regulation of T cell-mediated immunity

As an example, the interplay between metabolites and proteins for the immune process of positive regulation of T cell-mediated immunity is demonstrated (GO:0002711, Fig 7). The interaction network that was built for this process shows one big cluster of proteins and metabolites, and some unconnected proteins (not shown). Unconnected proteins may interact with non-immune-related proteins which are not considered in the current IMA methodology and indicate the current knowledge gap. Of the 8 metabolites in the network, 5 were found to be significant for the immune process (p<0.05, Table 2). These molecules are highly related as they interact with the same proteins. This results in the same centrality value for each molecule; however, the precision varies as they can interact with proteins in other processes. The exception here is 3-Dehydrospinganine, which only interacts with proteins involved in positive regulation of T cell-mediated immunity, resulting in a precision of 1.00.

Several significant metabolites form a primary component for sphingolipids. Sphingolipids are membrane lipids that function as ligands for sphingosine-1-phosphate receptors (S1PR) and are especially associated with the determination of T cell phenotypes [16]. Previous studies have shown that deficiency in S1PR can cause failure in mature T cells leaving the thymus

**Table 2. Metabolites associated with positive regulation of T cell-mediated immunity.** Fisher's exact test with FDR multiple testing correction was used to calculate p-values, while centrality and precision values are indicators of the importance of the metabolites in this process. The metabolite number in the table corresponds to the number in the interaction network in Fig 7.

| Metabolite | Metabolite superclass | Centrality | Precision | P-value | Metabolite number in network |
|---|---|---|---|---|---|
| **3-Dehydrosphinganine** | **Organic oxygen compounds** | **0.30** | **0.63** | **< 0.001** | **6** |
| **Phytosphingosine** | **Organic nitrogen compounds** | **0.30** | **0.56** | **< 0.001** | **7** |
| **Sphinganine** | **Organic nitrogen compounds** | **0.30** | **0.45** | **< 0.001** | **2** |
| **Sphingosine** | **Organic nitrogen compounds** | **0.30** | **0.39** | **< 0.001** | **1** |
| **Dihydroceramide** | **Organic acids and derivatives** | **0.30** | **0.31** | **< 0.001** | **8** |
| S-Adenosylmethionine | Nucleosides, nucleotides, and analogues | 0.22 | 0.07 | 1.00 | 4 |
| ADP | Nucleosides, nucleotides, and analogues | 0.22 | 0.01 | 0.58 | 5 |
| ATP | Nucleosides, nucleotides, and analogues | 0.22 | 0.01 | 0.58 | 3 |

[17]. Finally, it has been shown to be an important factor for coordinating adaptive immune responses through the $S1P_1$-Akt-mTOR pathway [18].

## IMatlas R package and R shiny application

We have implemented the IMA as an R package and R shiny module. The IMA supports several search modes to facilitate the construction of networks, using either immune processes or metabolites as input (Fig 8). An interaction network is built and evaluation metrics such as metabolite centrality and p-value are calculated. The IMA supports bulk input of HMDB identifiers or GO terms to produce graphs that can be adjusted using several thresholds using the settings panel. For example, confidence thresholds used by STRING for protein-protein interactions can be increased to include only very well-curated interactions. Other features include generating neighborhood graphs of a given set of metabolites and the ability to search using (super)classes and/or biochemical pathways. In summary, the application contains useful features to construct network graphs for non-programmatic applications.

## Discussion

We describe the development of the Immunometabolic Atlas (IMA), which leverages protein-metabolite interaction analysis to identify metabolites associated with immune processes, and vice versa, and which can be used to interpret and design metabolomics studies.

The IMA is based on metabolites included on the Human Metabolome Database (HMDB), which is a large, comprehensive, and well-annotated database of metabolites found in humans, and is more complete than alternative human metabolite databases [19]. HMDB contains a large number of lipid metabolites, which were found to be associated with many immune processes. Lipids are highly biologically relevant in various biological functions as is also extensively studied within the field of lipidomics [20, 21]. In this study, the superclass of lipids and lipid-like molecules was excluded from the validation and the example shown because the complex nomenclature of these metabolites made text mining unfeasible. However, inclusion of the superclass lipids and lipid-like molecules is available for researchers using the IMA. Not all lipids were removed by excluding this superclass. For example, sphingolipids were included in the example in Fig 7. Depending on the method of classification that is used, some lipid metabolites will be classified as such, and some will be classified further into other categories. The method of classification of metabolites by HMDB could be debated but is considered to be out of the scope of this study.

Association between metabolites and immune processes was based on the inferred protein-metabolite network, where proteins were associated with GO-associated immune processes

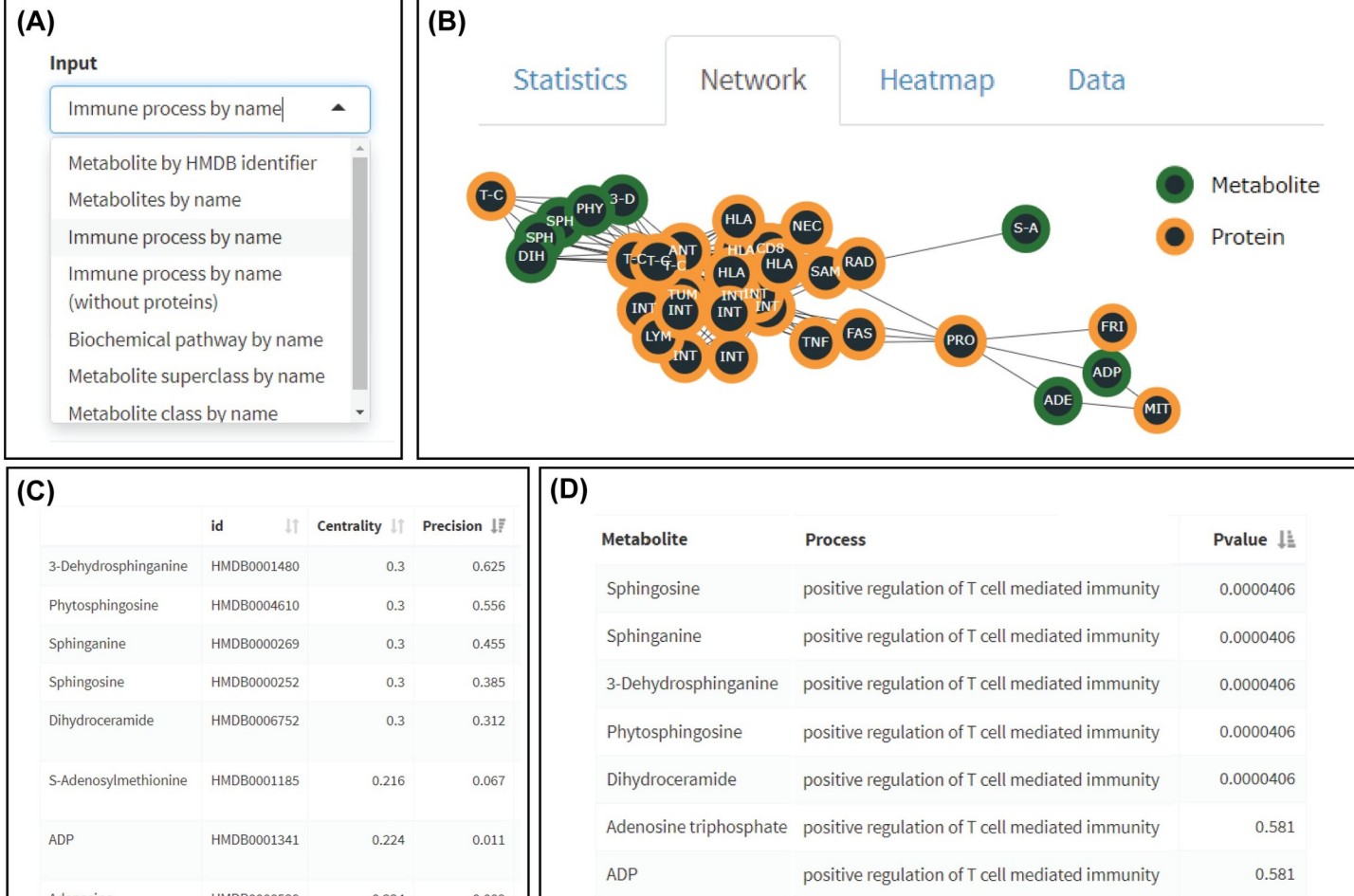

**Fig 8. IMA R shiny application concept.** (A) The application allows users to enter processes, metabolites, proteins, or identifiers from a list to produce networks and calculate statistics. This flexibility of input possibilities enables to obtain metabolites from immune processes and vice versa. (B) Several visualizations have been included to visualize associations between metabolites and processes. Here, we zoomed in on the connected part of the interaction network of T cell-mediated immunity. (C, D) Example outputs for metabolites associated with positive regulation of T cell-mediated immunity including centrality, precision, and p-values.

either through direct or higher-order inheritance based on protein-protein interactions present. The rationale for this strategy is based on the assumption that metabolites and proteins commonly interact: e.g., as enzyme-substrate or co-factor [22]. Of course, the majority of proteins are not limited to a single biological process. Evaluation of association strength of metabolite-immune processes through classical over-representation analysis is important to identify those metabolites or immune processes of primary interest [23]. Of note, the absolute p-value obtained from the over-representation analysis should be interpreted with caution because in the IMA not the whole metabolome for all human biological processes are used as a background, but only the metabolites and immune system processes in the IMA database.

To further interpret metabolites for their value as specific and selective biomarkers or drug targets we have included computation of precision and centrality network-structure inferred metrics. Centrality concerns the position of a metabolite in a network and has been proposed before [24]. The precision score assesses how specific the metabolite is for the associated process within the network. These metrics may help to identify metabolites of increased interest

as a biomarker for the specific immune process, e.g. because they are more likely to have a specific function, as inferred from the underlying network structure.

We evaluated immune response-metabolite associations through a comparison of literature text mining derived metabolite-immune process associations. We found that a substantial part (67%) of the IMA associations overlapped with the associations found in literature. Overall, the text mining approach identified a two times higher number of associations than the IMA. We expect that this is related to the nature of methodology used to identify associations, because of its intrinsic high likelihood of identifying false-positive associations, which we attempted to reduce through applying several filtering steps. Ultimately we found that the IMA yields a specificity of 73–99% and sensitivity of 11–49% depending on the inheritance method used, which indicates our method shows sufficient performance to be used as a tool for hypothesis generation or to guide metabolomics study design.

A similar tool for the functional interpretation of metabolomics study results is STITCH [6], which is a database incorporating known and predicted interactions between metabolites and proteins. STITCH assigns processes to metabolites based on direct interactions and a clustering-based algorithm. STITCH does not include topological measurements, whereas in the IMA this is applied for easier interpretation of larger networks. In contrast to the IMA, associations between GO biological processes and metabolites can only be made in the direction of metabolites to processes, but not from (immune) processes to metabolites.

Limitations of the current IMA include the lack of directionality of associations in the protein-metabolite network, which could help in identifying biochemical interactions that are most relevant and plausible. In addition, incorporation of data on cell-type-specific as well as (sub-) cellular locations of metabolites or metabolite-protein associations may help in refining metabolite-immune system associations inferred, in particular, because of the complex and multi-cellular nature of the immune response.

We conclude that the developed IMA can be a relevant tool to guide researchers in the field of immunometabolomics in the interpretation of immune-metabolomics data from experiments or clinical studies and to guide the design of prospective metabolomics studies in the field of immunology, which we facilitate by making our tool available both as R package and user-friendly web-application. Finally, we expect that the conceptual approach and developed algorithms for inferring metabolite-immune process associations through protein-metabolite interaction networks can be expanded towards complete biological ontologies, and is not just limited to immune processes.

## Supporting information

**S1 Table. Contents of IMA database.**
(DOCX)

**S2 Table. Summary of superclass characteristics in the IMA.**
(DOCX)

**S3 Table. Text mining results listing papers that mention both metabolite and immune process that is present in the IMA database.**
(CSV)

**S4 Table. All significant metabolite-immune process associations in the IMA.**
(CSV)

## Author Contributions

**Conceptualization:** Pascal Maas, Ilona den Hartog, Coen van Hasselt.

**Data curation:** Pascal Maas, Ilona den Hartog, Sonja Boman.

**Formal analysis:** Pascal Maas, Ilona den Hartog, Sonja Boman, Coen van Hasselt.

**Investigation:** Pascal Maas, Ilona den Hartog, Coen van Hasselt.

**Methodology:** Pascal Maas, Ilona den Hartog, Alida Kindt, Sonja Boman, Coen van Hasselt.

**Project administration:** Ilona den Hartog, Coen van Hasselt.

**Resources:** Coen van Hasselt.

**Supervision:** Ilona den Hartog, Thomas Hankemeier, Coen van Hasselt.

**Validation:** Pascal Maas, Ilona den Hartog, Sonja Boman, Coen van Hasselt.

**Visualization:** Pascal Maas, Ilona den Hartog, Coen van Hasselt.

**Writing – original draft:** Pascal Maas, Ilona den Hartog.

**Writing – review & editing:** Pascal Maas, Ilona den Hartog, Alida Kindt, Sonja Boman, Thomas Hankemeier, Coen van Hasselt.

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
