## [Decision Letter · Decision Letter 0]

21 Jan 2022

PONE-D-21-36250The Immunometabolic Atlas: a tool for design and interpretation of metabolomics studies in immunologyPLOS ONE

Dear Dr. den Hartog,

Thank you for submitting your manuscript to PLOS ONE. After careful consideration, we feel that it has merit but does not fully meet PLOS ONE’s publication criteria as it currently stands. Therefore, we invite you to submit a revised version of the manuscript that addresses the points raised during the review process.

When revising your manuscript, please take the comments of the reviewer into account, in particular those labeled "Major concerns" with the role of lipids and the connection to your research results, being the central points of concern.

We look forward to receiving your revised manuscript.

Kind regards,

Enrique Hernandez-Lemus, Ph.D.

Academic Editor

PLOS ONE

Journal Requirements:

3. Please include captions for your Supporting Information files at the end of your manuscript, and update any in-text citations to match accordingly. Please see our Supporting Information. 

Reviewers' comments:

Reviewer's Responses to Questions

**Comments to the Author**

1. Is the manuscript technically sound, and do the data support the conclusions?

Reviewer #1: Yes

2. Has the statistical analysis been performed appropriately and rigorously? 

Reviewer #1: Yes

3. Have the authors made all data underlying the findings in their manuscript fully available?

Reviewer #1: Yes

4. Is the manuscript presented in an intelligible fashion and written in standard English?

Reviewer #1: Yes

5. Review Comments to the Author

Reviewer #1: In their article entitled "The Immunometabolic Atlas: a tool for design and interpretation of metabolomics studies in immunology" the authors combine a series of databases and retrieve GO terms from Gene Ontology related to Immune processes to build a Immunometabolic network whose edges are defined by protein-protein, metabolite-protein and metabolite-metabolite interactions. Their IMA is compared to a bronze standard extracted by using a text mining strategy, a good idea that gives a sense of the space explored, but not much more assurance on the reliability of the approach (specially related to the exclusion of lipids). Fig 6 best summarizes the results and utility of the IMA as users could easily browse for high precision/highly central metabolites as these could be biomarkers for the associated immune processes.

Major concerns:

One of the big methodological decisions was to exclude lipids for obscure reasons such as:" Although this may illustrate and confirm the biological relevance of lipid metabolism in various biological functions [20,21], it also obscures the role and identification of non-lipid metabolites associated with immune processes. Furthermore, literature confirmation of lipid-immune process associations was found to be challenging due to nomenclature issues. "

But then why is the main example centered on the role of Sphingolipids in T-cell survival. Isn't this a lipid? Table 2 also shows basically lipids.

Are all the high precise/well connected metabolites related to lipids? If so the analysis of the authors becomes irrelevant and lipids need to be included back.

Minor concerns:

Fig 4A was very blurry, impossible to parse.

Although the manuscript is well written some repetitive and non-sensical phrase appear:

L 252 "Metabolites with only metabolite-metabolite interactions could not be

detected by the IMA because immune processes are only inherited through proteins in the current IMA methodology.The metabolites with protein-protein interactions were interacting with proteins that were not considered to be part of

an immune system process according to Gene Ontology and therefore not included in the IMA database. "

And some comments that seem to tautological: "We expect that these metrics can help identify and select metabolites

of increased interest as a biomarker for the specific immune process, e.g. because they are more likely to have a specific

function, as inferred from the underlying network structure."

Typo on L352, should be "processes" not "processed"

6. PLOS authors have the option to publish the peer review history of their article (what does this mean?). If published, this will include your full peer review and any attached files.

Reviewer #1: No

---

## [Author Response · Author response to Decision Letter 0]

26 Apr 2022

Reviewer #1

Major concerns:

One of the big methodological decisions was to exclude lipids for obscure reasons such as:" Although this may illustrate and confirm the biological relevance of lipid metabolism in various biological functions [20,21], it also obscures the role and identification of non-lipid metabolites associated with immune processes. Furthermore, literature confirmation of lipid-immune process associations was found to be challenging due to nomenclature issues. “ But then why is the main example centered on the role of Sphingolipids in T-cell survival. Isn't this a lipid? Table 2 also shows basically lipids.

Response: Lipids have not been excluded from the IMA tool described in this paper. Our IMA tool can be applied to explore lipid-immune process interactions. However, in the manuscript, we excluded the HMDB superclass lipids and lipid-like molecules for (1) the text-mining-based validation, and (2) the example analyses described. The reasons for doing so are: (1) lipids have a highly complex nomenclature which did not allow to apply the text-mining validation; (2) the much larger number of lipids in HMDB obscured interactions between other metabolite classes we intended to demonstrate in the example analyses of this manuscript. We adjusted the wording in the manuscript to clarify this: 

“We excluded the superclass lipids and lipid-like molecules as defined within HMDB from the validation because the complex nomenclature of these metabolites made text mining unfeasible.” (L161-162). 

We have not manually assigned metabolites to classes, but rely on the classification as defined within HMDB. Unfortunately, the HMDB classification of metabolites still includes some lipids, even after excluding the entire HMDB metabolite superclass lipids and lipid-like molecules. Differences in metabolite classification schemes are beyond the scope of this manuscript. We clarified this in the discussion of the manuscript: 

“HMDB contains a large number of lipid metabolites, which were found to be associated with many immune processes. Lipids are highly biologically relevant in various biological functions as is also extensively studied within the field of lipidomics [20,21]. In this study, the superclass of lipids and lipid-like molecules was excluded from the validation and the example shown because the complex nomenclature of these metabolites made text mining unfeasible. However, inclusion of the superclass lipids and lipid-like molecules is available for researchers using the IMA. Not all lipids were removed by excluding this superclass. For example, sphingolipids were included in the example in Figure 7. Depending on the method of classification that is used, some lipid metabolites will be classified as such, and some will be classified further into other categories. The method of classification of metabolites by HMDB could be debated, but is considered to be out of the scope of this study.” (L348 – 356)

The HMDB classification of lipids to other non-lipid superclasses is also the reason why in Table 2 some lipids remain, even after excluding the lipid superclass. We have added a column stating the HMDB metabolite superclass to table 2 to clarify their classification and added a sentence in the discussion to clarify: 

“Not all lipids were removed by excluding this superclass. For example, sphingolipids were included in the example in Figure 7.” (L353-354)

Are all the high precise/well connected metabolites related to lipids? If so the analysis of the authors becomes irrelevant and lipids need to be included back.

Response: The high precise / well connected metabolites are relatively evenly distributed over all metabolite superclasses. We have added a new supplementary table (S2 Table) displaying a summary of superclass characteristics to visualize this. Also, we added to L226-227: 

“The average number of immune processes per protein is in the same order of magnitude for all superclasses except for benzenoids”. 

Combining the information in Tables S1 and S2, the reviewer can now also find that the superclass of lipids and lipid-like molecules interacts with 5% of proteins, while simultaneously involving 93% of all metabolites in the database. This and a possible explanation is also stated in the manuscript: 

“Metabolites of the superclass of lipids and lipid-like molecules, here referred to as ‘lipids’, were abundantly present with 90 280 occurrences (92.6%) but interacted with a relatively small portion of proteins (5.1%).” (L227-229)

Fig 4A was very blurry, impossible to parse.

Response: We have uploaded Figure 4 as a high-resolution file. It can be downloaded using the link referring to the original file. 

Although the manuscript is well written some repetitive and non-sensical phrase appear.

L 252 "Metabolites with only metabolite-metabolite interactions could not be detected by the IMA because immune processes are only inherited through proteins in the current IMA methodology. The metabolites with protein-protein interactions were interacting with proteins that were not considered to be part of an immune system process according to Gene Ontology and therefore not included in the IMA database. "

Revised as follows: “Metabolites that were only interacting with other metabolites, and not with proteins, could not be detected because immune processes are only inherited through proteins in the current IMA methodology. Metabolites that were only interacting with proteins that were not in the immune system process (according to GO), were also not included in the IMA database.” (L255-258).

And some comments that seem to tautological: "We expect that these metrics can help identify and select metabolites of increased interest as a biomarker for the specific immune process, e.g. because they are more likely to have a specific function, as inferred from the underlying network structure."

This was rewritten to: “These metrics may help to identify metabolites of increased interest as a biomarker for the specific immune process, e.g. because they are more likely to have a specific function, as inferred from the underlying network structure.” (L369-371)

Typo on L352, should be "processes" not "processed"

Changed.

We have now performed a full review of the writing with additional minor corrections.

---

## [Decision Letter · Decision Letter 1]

29 Apr 2022

The Immunometabolic Atlas: a tool for design and interpretation of metabolomics studies in immunology

PONE-D-21-36250R1

Dear Dr. den Hartog,

We’re pleased to inform you that your manuscript has been judged scientifically suitable for publication and will be formally accepted for publication once it meets all outstanding technical requirements.

Kind regards,

Enrique Hernandez-Lemus, Ph.D.

Academic Editor

PLOS ONE

Additional Editor Comments (optional):

Reviewers' comments:

Reviewer's Responses to Questions

**Comments to the Author**

1. If the authors have adequately addressed your comments raised in a previous round of review and you feel that this manuscript is now acceptable for publication, you may indicate that here to bypass the “Comments to the Author” section, enter your conflict of interest statement in the “Confidential to Editor” section, and submit your "Accept" recommendation.

Reviewer #1: All comments have been addressed

2. Is the manuscript technically sound, and do the data support the conclusions?

Reviewer #1: Yes

3. Has the statistical analysis been performed appropriately and rigorously? 

Reviewer #1: Yes

4. Have the authors made all data underlying the findings in their manuscript fully available?

Reviewer #1: Yes

5. Is the manuscript presented in an intelligible fashion and written in standard English?

Reviewer #1: Yes

6. Review Comments to the Author

Reviewer #1: All comments have been addressed, there is nothing more to add, the manuscript can be now evaluated by the editor.

7. PLOS authors have the option to publish the peer review history of their article (what does this mean?). If published, this will include your full peer review and any attached files.

Reviewer #1: No

---

## [Editor Report · Acceptance letter]

4 May 2022

PONE-D-21-36250R1 

The Immunometabolic Atlas: a tool for design and interpretation of metabolomics studies in immunology 

Dear Dr. den Hartog:

I'm pleased to inform you that your manuscript has been deemed suitable for publication in PLOS ONE. Congratulations! Your manuscript is now with our production department. 

Kind regards, 

on behalf of

Prof. Enrique Hernandez-Lemus 

Academic Editor

PLOS ONE